# Learning Invariance Manifolds of Visual Sensory Neurons

**Luca Baroni**[†]                                    BARONI@KSVI.MFF.CUNI.CZ
*Faculty of Mathematics and Physics, Charles University, Prague, Czechia*

**Mohammad Bashiri**[†]                    MOHAMMAD.BASHIRI@UNI-TUEBINGEN.DE
*Institute for Bioinformatics and Medical Informatics, University of Tübingen, Germany*

**Konstantin F. Willeke**          KONSTANTIN-FRIEDRICH.WILLEKE@UNI-TUEBINGEN.DE
*Institute for Bioinformatics and Medical Informatics, University of Tübingen, Germany*

**Ján Antolík**                                    ANTOLIKJAN@GMAIL.COM
*Faculty of Mathematics and Physics, Charles University, Prague, Czechia*

**Fabian H. Sinz**                              SINZ@CS.UNI-GOETTINGEN.DE
*Campus Institute Data Science, University Göttingen, Germany*
*Institute for Bioinformatics and Medical Informatics, University of Tübingen, Germany*

[†]*denotes equal contribution*

**Editors:** Sophia Sanborn, Christian Shewmake, Simone Azeglio, Arianna Di Bernardo, Nina Miolane

## Abstract

Robust object recognition is thought to rely on neural mechanisms that are selective to complex stimulus features while being invariant to others (e.g., spatial location or orientation). To better understand biological vision, it is thus crucial to characterize which features neurons in different visual areas are selective or invariant to. In the past, invariances have commonly been identified by presenting carefully selected hypothesis-driven stimuli which rely on the intuition of the researcher. One example is the discovery of phase invariance in V1 complex cells. However, to identify novel invariances, a data-driven approach is more desirable. Here, we present a method that, combined with a predictive model of neural responses, learns a manifold in the stimulus space along which a target neuron's response is invariant. Our approach is fully data-driven, allowing the discovery of novel neural invariances, and enables scientists to generate and experiment with novel stimuli along the invariance manifold. We test our method on Gabor-based neuron models as well as on a neural network fitted on macaque V1 responses and show that 1) it successfully identifies neural invariances, and 2) disentangles invariant directions in the stimulus space *.

**Keywords:** neural invariances, invariance manifold, MEI, disentanglement, contrastive learning, visual cortex, CPPN

## 1. Introduction

Visual sensory areas enable animals to identify objects robustly under different viewing conditions and contexts. Such ability is thought to require neural mechanisms that are selective to complex stimulus features but invariant to others (e.g., spatial location or rotation). To better understand biological vision, it is thus crucial to characterize which features strongly drive neural activity and identify which transformations of such features

---

*. Code is available at https://github.com/sinzlab/cppn_for_invariances.

leave neural responses unchanged – i.e. single cell invariances. In the past, identification of invariances in visual sensory systems have commonly been a hypothesis-driven process relying on presentation of carefully selected stimuli. One example of this is the discovery of phase invariance in complex cells of primary visual cortex (Hubel and Wiesel, 1962). However, such an approach heavily relies on the intuition of the experimenter or serendipity. Since the dimensionality of images is enormous and experimental time is limited, this approach quickly becomes infeasible when encoding of visual information becomes more complex in higher areas.

In recent years, artificial neural networks trained on large datasets of neural responses to natural images have proven to be powerful predictive models of neural responses (Yamins et al., 2014; Kriegeskorte, 2015; Antolík et al., 2016; Yamins and DiCarlo, 2016; Klindt et al., 2017; Cadena et al., 2019; Kubilius et al., 2019; Sinz et al., 2018; Lurz et al., 2021; Zhuang et al., 2021). An alternative approach might thus be to systematically explore the invariance space of visual sensory neurons via optimization using these predictive models. A large body of research in the field of interpretable machine learning has focused on feature visualization, a set of techniques to identify which inputs highly activate the network units or layers (Olah et al., 2017). These techniques have already been successfully used to find single (Walker et al., 2019; Bashivan et al., 2019; Ponce et al., 2019) or multiple (Cadena et al., 2018; Ding et al., 2022) maximally exciting stimuli for visual sensory neurons. However, all current methods predict only a discrete set of stimuli from the invariance manifold. Considering the high dimensionality of images, understanding how such stimuli are connected in the image space can be non-trivial, especially when neurons are invariant to multiple transformations, as it is expected to be more and more the case along the visual hierarchy.

Here, we present a systematic data-driven approach based on implicit image representations and contrastive learning, that allows the identification and parameterization of the manifold of highly activating stimuli. We refer to this manifold as MEI invariance manifold (or just invariance manifold for simplicity). We first tested our method on simple Gabor-based toy models that exhibit multiple invariances and different invariant manifold topologies. We found that our method correctly identifies and disentangles different invariance directions. We then validated our method on selected macaque V1 neurons where it identifies an almost exact phase invariance. Taken together, our results show that our approach can capture invariance manifolds in a meaningful way and can be potentially used to discover novel invariances in visual sensory neurons.

## 2. Related work

**Most Exciting Image (MEI) via pixel optimization**   Artificial neural networks have been recently used to synthesize images that maximize the response of a given neuron in the visual system of mice and monkeys (Walker et al., 2019; Bashivan et al., 2019; Ponce et al., 2019). Such MEIs were commonly identified via direct optimization of pixel values. This is a well established technique in the field of interpretable machine learning for inspecting the units and their function in artificial neural networks (Erhan et al., 2009; Olah et al., 2017). Importantly, Walker et al. (2019); Bashivan et al. (2019); Ponce et al. (2019) demonstrated that these MEIs indeed activate biological neurons stronger than control stimuli, such as Gabors, in most cases. These results thus demonstrate the utility of these models as digital

twins of the biological brain, allowing neuroscientists to conduct analyses *in-silico* that are infeasible to perform on the biological system, but whose predictions can be verified *in-vivo*.

**Diverse feature visualization** Previous works have mostly focused on identifying a single MEI for a single (Walker et al., 2019) or a population of neurons (Bashivan et al., 2019; Ponce et al., 2019). However, it is not clear whether there exist only a single MEI or rather a manifold of maximally exciting images. To inspect the presence of such invariances, Cadena et al. (2018) expanded on the same technique, optimizing for multiple images (diverse MEIs) while enforcing diversity with an additional objective. Such an approach allows the identification of multiple distant points in the manifold of maximally exciting images. Given that the space of images is very high dimensional, the question remains how to connect such points to construct an invariance manifold. For instance, different phases of an optimal Gabor stimulus of a complex cell cannot be connected by straight lines in image space. The mid point between two 180 degree shifted Gabors would be a flat image, which is certainly not strongly driving a complex cell. Instead, the maximally exciting curve between the two Gabors forms a circle in high dimensions.

**Differentiable Image Parameterization** Recent developments in feature visualization techniques show that smooth, semantically meaningful, transition between images can be obtained via differentiable parameterization methods (Mordvintsev et al., 2018; Ha, 2016; Mildenhall et al., 2021). Such methods are, however, yet to be applied to characterize the invariances of biological neurons.

## 3. Methods

In contrast to previous approaches to identify MEIs, i.e. directly optimizing pixel values, we use Compositional Pattern Producing Networks (CPPNs) to optimize a reparameterized version of the image. CPPNs (Stanley, 2007) are artificial neural networks mapping pixel positions $(x, y)$ to pixel RGB (or grayscale) values. They have recently gained a lot of attention in the computer vision community as implicit representations of shapes and radiance fields (Ha, 2016; Mescheder et al., 2019; Mildenhall et al., 2021). A vanilla CPPN is a differentiable implicit representations of a single image in arbitrary resolution.

### 3.1. CPPN as an implicit representation of the invariance manifold

Our goal is to use a single CPPN as an implicit representation of not a single image but the whole manifold of images that equally maximize the activation of a target neuron. For this, a single CPPN needs to produce a variety of images. This can be achieved by extending the inputs of the CPPN to include an additional input variable $z$ belonging to a low-dimensional bounded latent space. This allows the CPPN to output different images while being fed the same set of pixel positions (Ha, 2016). In the context of learning the invariance manifold, different values of $z$ should result in different images that maximally excite a target neuron. If this is achieved, $z$ captures a latent parameterization of the MEI invariance manifold and a specific value of it represents a single point on the manifold. We implemented the CPPN as a simple fully-connected neural network of 8 hidden layers each with 15 units. Each hidden layer was followed by a batch normalization and leaky ReLU nonlinearity. As we considered only grayscale images, the output layer of the CPPN contains a single unit with a

BARONI† BASHIRI† WILLEKE ANTOLÍK SINZ

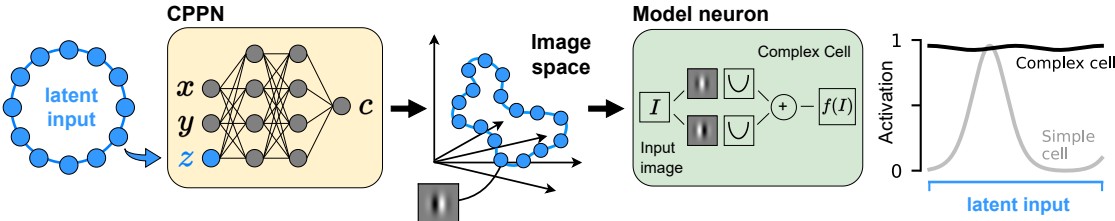

Figure 1: Our method uses a CPPN to map a simple low-dimensional latent space onto a complex high-dimensional manifold in the image space. Images from this manifold result in diverse but maximally exciting stimuli for a model neuron. Here we show a schematic of this method applied on a complex cell. Corresponding activations for a simple cell are also added as reference.

tanh nonlinearity, resulting in a 1D output with values between -1 and 1. To allow control over the characteristic spatial frequency of the patterns generated via CPPN, instead of directly using pixel positions as inputs to the CPPN, we used positional encoding of the pixel positions via random Fourier mapping (Tancik et al., 2020; Mildenhall et al., 2021)[1].

As the topology of the MEI invariance manifold might vary from neuron to neuron, we explored different dimensionalities and boundary conditions for the latent space ("latent input" in Fig. 1). In particular, we considered 1D and 2D latent spaces, non-periodic (corresponding to a line or sheet topology) or periodic (corresponding to circle or torus topology).

### 3.2. Constrastive objective for image diversification

After the CPPN maps the low-dimensional latent to a manifold in image space, these images are fed to a predictive model of neural responses (Fig. 1). The parameters of the CPPN are then optimized to (i) maximally excite a target model neuron and (ii) produce diverse images. To enforce the latter objective, we train the CPPN with a contrastive objective function (Chopra et al., 2005) which encourages image diversification. Specifically, for each point $z_i \in \mathbb{R}^D$, belonging to a grid of values covering the $D$-dimensional latent space, the objective function to be maximized is composed of two terms:

$$\mathcal{L} = \mathcal{L}_{\text{act}} + \mathcal{L}_{\text{contrastive}} \tag{1}$$

The first term $\mathcal{L}_{\text{act}}$ represents the resulting neural activation from the generated image $I(z_i)$, and encourages the CPPN to generate images that highly activate the neuron:

$$\mathcal{L}_{\text{act}} = \frac{\alpha_i}{\alpha_{MEI}},$$

---

1. Each position $(x, y)$ gets mapped to a $k$-dimensional space followed by $\sin(\cdot)$ and $\cos(\cdot)$ transformations: $[\sin(\mathbf{b}[x, y]^\top), \cos(\mathbf{b}[x, y]^\top)]$ where $\mathbf{b} \sim \mathcal{N}(\mathbf{0}, \sigma\mathbf{I})$ is randomly sampled from a $k$-dimensional normal distribution. Here, we used $k = 10$ and $\sigma = 1$.

where $\alpha_i$ is the model neuron's response to image $I(z_i)$ and $\alpha_{\text{MEI}}$ is the neuron's MEI activation obtained through standard pixel optimization (see Appendix A for implementation details of MEI generation via pixel optimization). The normalization by the neuron's MEI activation results in a maximal objective value around 1. The second term $\mathcal{L}_{\text{contrastive}}$ is based on soft nearest neighbor contrastive objective (Salakhutdinov and Hinton, 2007; Frosst et al., 2019). It uses positive and negative images to encourage a manifold of generated images to expand and be meaningfully parameterized by the latent coordinates:

$$\mathcal{L}_{\text{contrastive}} = c \cdot \log \frac{\frac{1}{N_+} \sum_{z_j \in \mathcal{Z}_+} \exp(\text{sim}(I(z_i), I(z_j))/\tau)}{\frac{1}{N_-} \sum_{z_k \in \mathcal{Z}_-} \exp(\text{sim}(I(z_i), I(z_k))/\tau)}. \tag{2}$$

Specifically, for each latent grid point $z_i$ a set of "positive" neighboring points $\mathcal{Z}_+$ is defined on the grid. The rest of the grid points that are further from $z_i$ are treated as "negative" points and are denoted as $\mathcal{Z}_-$. We use cosine similarity as a similarity measure on the corresponding generated images. The numerator of the logarithm in Eq. (2) thus enforces images corresponding to close-by points to look similar, while the denominator forces images corresponding to distant points to look different. A temperature parameter $\tau$ regularizes this term (Wang and Liu, 2021) to control the diversity of images generated by the CPPN. We also used a scaling factor $c$ to control the strength of the $\mathcal{L}_{\text{contrastive}}$ contribution to the full objective in Eq. (1). Finally, we average the single terms given by Eq. (1) across all grid points resulting in the complete objective function to maximize during training:

$$\mathcal{L} = \frac{1}{N^D} \sum_{z_i \in \mathcal{Z}} (\mathcal{L}_{\text{act}} + \mathcal{L}_{\text{contrastive}}). \tag{3}$$

Here, $N^D$ denotes the total number of grid points, $D$ the number of latent dimensions, and $N$ the number of grid points per dimension.

### 3.3. Training the CPPN

At each step, a grid of $N^D$ evenly spaced points covering values between 0 and $2\pi$ (in each dimension) is constructed in the latent space. To allow the CPPN to learn meaningful representations not only at discrete positions, but on the whole latent space, a random jitter $\epsilon \in [-\frac{a}{2}, \frac{a}{2}]^D$ is added to the entire grid, where $a$ is the spacing between grid points in each latent dimension. If required, periodicity on the latent space is enforced by applying $\sin(\cdot)$ and $\cos(\cdot)$ functions on the grid points before passing them to the CPPN (i.e. $z \to [\cos(z), \sin(z)]$). The CPPN generates a grid of images corresponding to the latent grid points. Subsequently these images are rescaled to have a fixed mean (luminance) and standard deviation (contrast) and passed to the ANN model predicting neural activation. The constraint on the luminance and contrast allows for the comparison between the responses across multiple images and forces highly driving features to appear in the receptive field of the neuron, while flattening the rest of the image (for training details refer to Appendix A).

### 3.4. Predicting neural responses of macaque V1

**Neuronal data**  The neuronal data have been described previously in (Cadena et al., 2022). In brief, responses of neurons in medial primary visual cortex at eccentricities ranging

from 1.4 to 3.0 degrees of visual angle were recorded from two rhesus macaque monkeys. Using 32-channel linear silicon probes, a total of 458 neurons were isolated in 15 (monkey 1) and 17 (monkey 2) sessions. Neural activity was recorded in response to natural images from ImageNet (Deng et al., 2009) while the monkeys were fixating on a central fixation spot. Each image was shown for 120ms, and spikes were extracted from 40 to 160 ms after image onset. Per recording session, between 10,000 and 15,000 unique images out of a pool of 24075 ImageNet images were presented in blocks of 15. All images were displayed in grayscale, with a resolution of 63 pixels per degree (ppd), covering 6.7 degrees visual angle on the monitor.

**Predictive model** The artificial neural network (ANN) model we used for predicting neural responses from macaque V1 is inspired by previous deep network models (Cadena et al., 2019; Lurz et al., 2021). It consists of a nonlinear core which captures general image representations, and a readout that maps the core representations onto scalar neuronal responses via regularized regression. As core, we used a CNN with depth separable convolutions (all layers except the first), with 3 layers and 32 feature channels per layer. After each convolutional layer, a batch normalization followed by an ELU nonlinearity are applied. From the last layer, a pyramid readout (Sinz et al., 2018) extracts the features at a learned spatial location $(x, y)$ as well as at the same location in two progressively downsampled versions of the last layer's output. We used average pooling with a kernel size of 3 in each downsampling step. $n = 96$ weights per neuron are then learned to linearly combine the features from the last layers and its two downsampled versions. The resulting outputs are then passed through an ELU + 1 nonlinearity to finally obtain the scalar positive firing rate for each neuron.

**Training the ANN on monkey V1 responses** We first cropped the images to the central 2.65 degrees (from the original 6.7 degrees) and subsequently downsampled the resolution to 35 pixels per degree, leading to an input size of $93 \times 93$ pixels for the ANN model. Prior to model training, we split all stimuli into 19200 training, 4800 validation, and 75 test images, and z-scored all images based on the mean and standard deviation across the training and validation set. We trained our ANN by minimizing the Poisson loss $\frac{1}{m} \sum_{i=1}^{m} \left( \hat{r}^{(i)} - r^{(i)} \log \hat{r}^{(i)} \right)$, where $m$ denotes the number of neurons, $r$ the observed neuronal firing rate, and $\hat{r}$ the predicted firing rate. We then optimized the parameters of the ANN using the Adam (Kingma and Ba, 2014) optimizer with a learning rate of 0.0042. We decreased the learning rate by a factor of 0.3 when the validation loss did not decrease for three consecutive epochs for a maximum of 3 times before stopping the training altogether.

## 4. Results

We tested our method on simple Gabor-based model neurons with known (and exact) invariances and on neural network models predicting the responses of macaque V1 neurons. On synthetic data, we tested our approach on model neurons with a variety of invariances. While the method can be applied to arbitrarily high-dimensional invariance manifolds, here we considered model neurons with 1D and 2D invariances to easily visualize the results and facilitate interpretation of the learned invariances. Specifically, we considered a simple cell

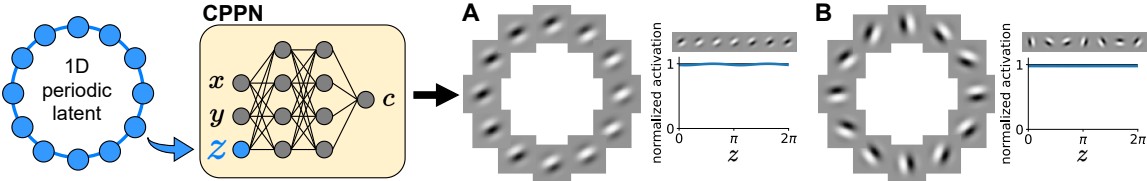

Figure 2: Invariances generated from equally spaced points in a periodic 1D latent space in the case of a complex cell (A) and of a orientation invariant neuron (B) and activation values to different images in the corresponding learned manifold

corresponding to a single point (i.e. no invariance), a complex cell (phase invariance) as well as an orientation-invariant neuron corresponding to a circle, and a phase-and-orientation-invariant neuron corresponding to a torus[2]. In the case of phase-and-orientation-invariant neuron, we additionally considered a partial orientation invariance covering only 90 degrees, resulting in a cylinder invariance topology (see Appendix B for implementation details). This variety of topologies allowed us to test how robustly our method parameterizes the entire invariance manifold, whether the parameterization associates meaningful directions to the axes of the latent space, and how it behaves when the topology of the latent space does not match the one of the invariance manifold.

**Learning invariance manifolds with 1D latent spaces** First, we explored how our method parameterizes 1D invariances in the case of complex and orientation-invariant model neurons. Since both phase and orientation represent periodic transformations, the invariance manifold of these cells have the topology of a circle. We therefore first tested a 1D periodic latent variable $z$ as input to the CPPN. Our method identified the invariance manifold almost perfectly (Fig. 2). Specifically, the latent space input represents a parameter that corresponds to the angle characterizing the invariance. Next, we considered a non-periodic 1D latent space – topology of a line. In this case, the temperature parameter of the contrastive objective seems to affect the extend of the invariance manifold captured by the CPPN. The reason for this behavior is a mismatch between the true invariance topology (a circle) and the fitted topology (a line): for a line topology, opposite boundaries in the latent space correspond to negative samples, and the contrastive objective encourages them to look dissimilar discouraging the model to complete a full circle (see Appendix C for a more thorough analysis). Nonetheless, even in this case, the generated invariance manifold well adheres to a part of the ground truth invariance manifold and achieves a meaningful parameterization of the invariance (see Fig. S2).

In the scenarios we considered so far the latent space dimensionality matched the dimensionality of the invariance manifold. However, it is possible that the underlying invariance manifold is higher-dimensional than the CPPN's latent space. To see how it behaves in such a scenario, we applied a CPPN with a 1D latent space on a phase-and-orientation-invariant neuron (2D invariance). While it is not possible to capture the complete invariance mani-

---

2. To be more precise, the MEI invariance manifold of a phase-and-orientation-invariant neuron has the topology of torus that touches itself for phases corresponding to even Gabors and rotations of 180 degrees.

BARONI[†] BASHIRI[†] WILLEKE ANTOLÍK SINZ

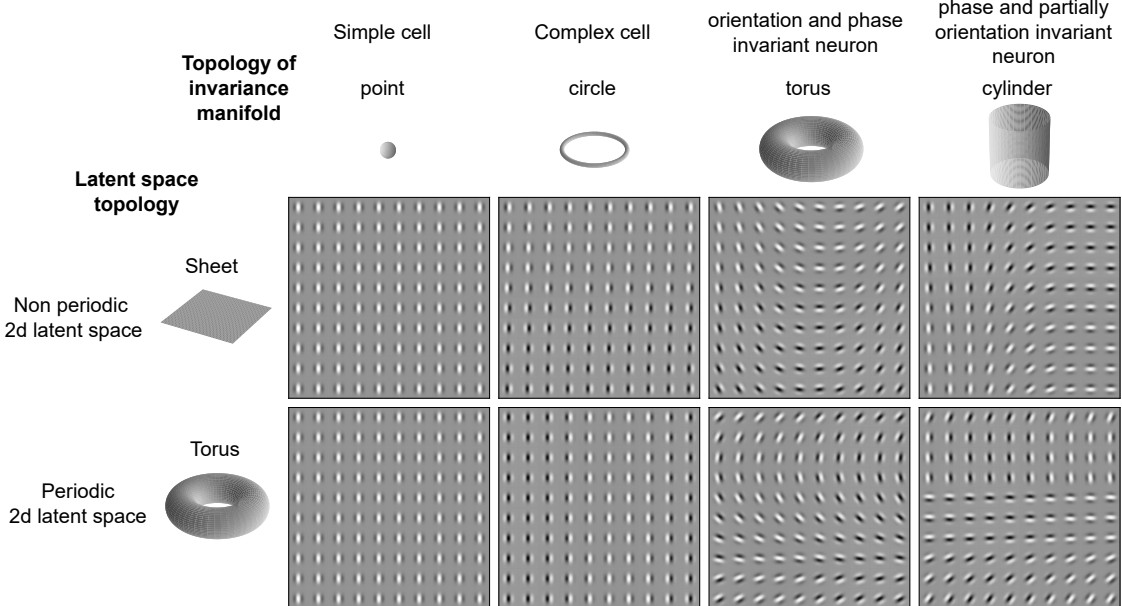

Figure 3: Invariances learned with with 2D latent space for different configurations of latent space topology and topologies of the ground truth invariance manifold. Mean and standard deviation of the activations corresponding to the shown images are reported in Appendix G.

fold in this case, in Appendix D we show that our method still learns a submanifold of the higher-dimensional invariance manifold.

Our method implicitly assumes the invariance manifold to be continuous and so far we tested it on smooth ground truth invariances. As a final test, we also assessed how well it can capture discontinuous manifolds. Our results (Fig. S5) show that our method can indeed learn to approximate discontinuous invariance manifolds successfully (for details refer to Appendix E).

**Learning invariance manifolds with 2D latent spaces** Subsequently, we considered a 2D latent space with non-periodic (sheet topology) and periodic (torus topology) invariances and trained a CPPN to identify the invariances of all the neuron models mentioned above (Fig. 3). In the case of a simple cell, the CPPN learned to ignore the invariance latent variable $z$ and collapsed the predicted invariance manifold onto a single point, matching the Gabor filter corresponding to the MEI of the cell (Fig. 3 left column). In the case of a complex cell, the CPPN learned to ignore one latent dimension and associated the invariance transformation with the other (Fig. 3 second column from left), as it would be expected in the ideal case. In a similar fashion to the 1D case (Fig. 2), the CPPN with non-periodic latent learned an incomplete yet meaningful parameterization of the invariance, whereas the CPPN with periodic latent learned the full invariance.

In the case of the jointly orientation-and-phase-invariant neuron, the CPPN learned both invariances and disentangled them in the latent space nearly perfectly (Fig. 3 second column

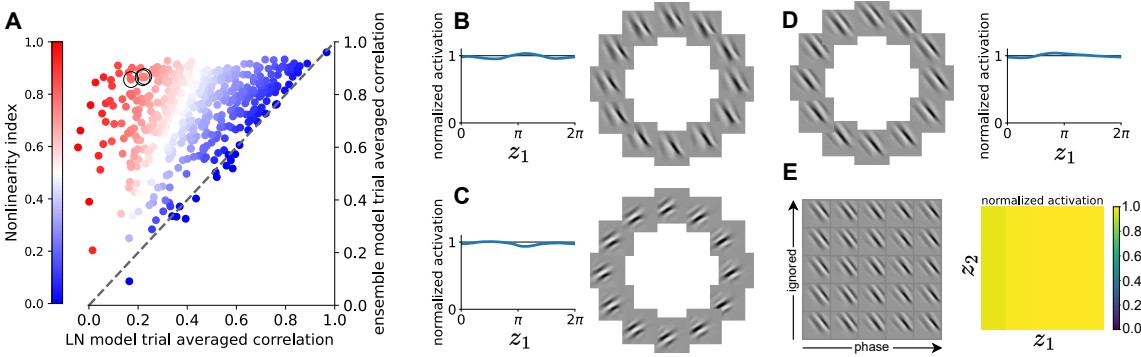

Figure 4: **A**: Nonlinearity index of macaque V1 neurons. Black circles highlight the neurons shown in panels **B**–**E**. **B**–**E**: Phase invariances identified with periodic 1D latent (**B**–**D**) and with 2D non-periodic latent(**E**) and corresponding activations. For visualization purposes, MEIs are cropped around the receptive field of the neurons.

from right). This result is particularly relevant, as it allows both clear identification of the invariance and control over it via the latent space. Again, the CPPN mapping from non-periodic latent space parameterized only half of the true periodic invariance transformations. We then explored the scenario of a phase-and-partially-orientation-invariant model neuron. The invariance manifold of this neuron model has the topology of a cylinder. In this case, the CPPN with a non-periodic latent space learned both invariance transformations and disentangled them along the two latent space dimensions (Fig. 3 right column top row). In line with the previous results, the partial orientation invariance transformation was learned fully, whereas the phase transformation was learned up to a 180 degree phase shift (Fig. 3 right column top row). Fitting a periodic latent space on a non-periodic invariance manifold topology is more complex. Specifically, to deal with non-periodic invariances, the CPPN mapping from periodic latent can either learn the full invariance transformation twice, or introduce sudden jumps in the invariance manifold (Fig. 3 right column bottom row). Our experiments indicate that both scenarios can happen (see Fig. S6) and that the CPPN can still learn to disentangle the two transformations.

**Learning the invariance manifold of macaque V1 complex cells**   Lastly, we set out to validate our method on a model for a population of macaque V1 neurons (see section 3.4). The Gabor-based model neurons above presented (almost) exact invariances, with no fluctuations over activation levels. In a biological neuron, however, a meaningful definition of the MEI invariance manifold should be more forgiving, allowing for a broader variety of images to be considered maximally exciting, for the following reasons: First, it is not to be expected that biological neurons present exact invariances over maximally exciting stimuli. Second, the data collected and analyzed in neurophysiological experiments are intrinsically noisy and limited in size. Third, our experiments here are performed on neural network models fitted to neural responses, which despite achieving high predictive performance, are not perfect. We used an ensemble model of ANNs (see section 3.4) as a model of of macaque V1 neurons and applied our method on complex cells that were identified using a nonlin-

earity index (Antolík et al., 2016). See Appendix H for details on selecting complex cells. Fig. 4 shows that the CPPN found phase invariance in the selected neurons: it generated a variety of maximally exciting images resembling Gabor filters and parameterized their phase transformation with one of the latent space dimension (see Appendix I for a more thorough analysis of the learned phase invariance). This demonstrates the ability of the method to identify invariances in biological neural representations.

## 5. Discussion

We presented a data-driven method that combines a CPPN with a contrastive learning objective to map from a low-dimensional latent space to a manifold in the space of images that describes the MEI invariances of a given neuron. We tested our approach on synthetic neural responses, where multiple ground truth exact invariance manifolds were known, as well as on predictive models of macaque V1 complex cells. We showed that our approach successfully uncovers MEI invariance manifolds in both scenarios. In contrast to previously presented approaches, our method allows a smooth parameterization of the invariance manifold and, when multiple invariances are present, it disentangles them along the axes of the latent space. When the dimensionality of the latent space is higher than the dimensionality of the invariance, the CPPN learns to ignore unnecessary latent dimensions.

In the future, our approach can be extended to learn implicit representation of MEI invariances for multiple neurons, for instance by associating to each neuron a learnable embedding used as a fingerprint. Such multi-neuron implementation, in combination with a regularization of the space of fingerprints, could, in principle, allow us to classify neurons in functional clusters, according to their invariances. Similarly, a multi-neuron implementation could allow us to study interesting tuning directions, i.e. directions in image space to which certain neurons are selective, whereas others are invariant. We believe that the approach presented here will prove to be a valuable tool to advance our understanding of visual sensory coding, especially in the higher visual areas, such as V4, that potentially exhibit more invariances, both in terms of quantity and complexity.

### Acknowledgments

We thank Alexander Ecker and the members of Sinz lab for their helpful feedback and discussions. The authors thank Tolias lab for providing recordings from macaque V1. We also thank the International Max Planck Research School for Intelligent Systems (IMPRS-IS) for funding MB and supporting KFW. This project has received funding from the European Union's Horizon 2020 research and innovation programme under the Maria Sklodowska-Curie grant agreement No 861423. FHS was supported by the Carl-Zeiss-Stiftung and acknowledges the support of the DFG Cluster of Excellence "Machine Learning – New Perspectives for Science", EXC 2064/1, project number 390727645.

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

## Appendix A. Additional training details

### A.1. CPPN training details

The CPPN was trained via gradient-based optimization to maximize the objective in Eq. 3. During training, the contrastive objective requires a set of positive and negative input images for each latent input. This was achieved with the construction of masks identifying positive and negative points. Positive neighboring areas are as squares surrounding the point considered, and extending from it in each direction for $0.1 * N$ number of points (Fig. S1). The periodicity condition of the latent space is reflected in the masks of points close to the boundaries. Regularization strengths were rescaled as $c = \bar{c} \times \frac{\tau}{2}$ to normalize the maximum possible contribution coming from a single point in the contrastive term depending on temperature. We observed that strong initial regularization seems to disentangle the invariant directions and to avoid sudden jumps in image space as a function of the latent space, but can concurrently deteriorate the MEIs generated. This is due to the fact that the objective in this case has to balance between maximizing activation and satisfying comparably strong regularization conditions. For this reason during training we decrease $\bar{c}$.

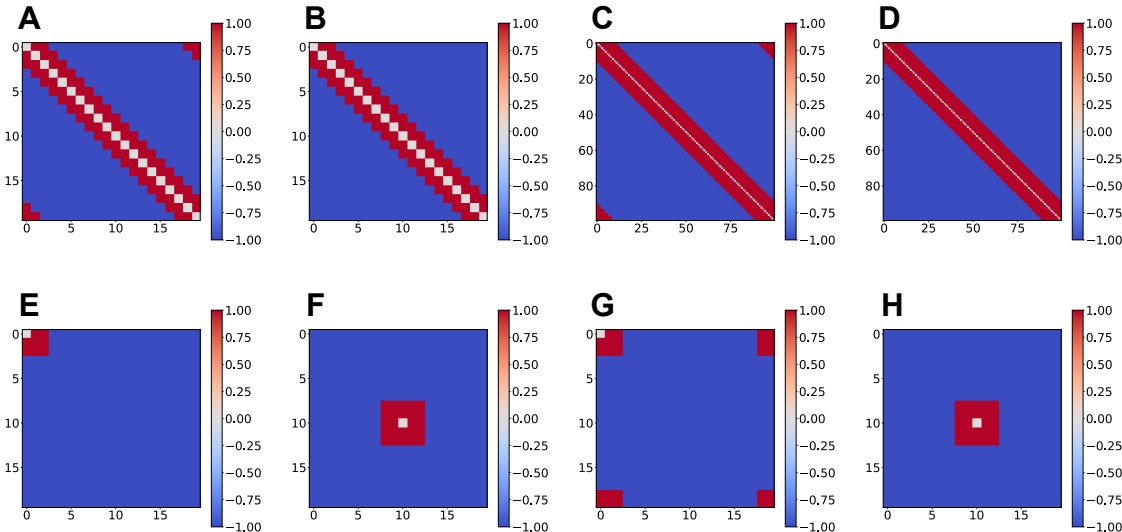

Fig. S1: Masks to determine positive (red) and negative (blue) examples on the grid for different conditions. Neighbouring size set to $s_n = 0.1$. (**A-D**): Masks for all points in 1D grid (rows of each matrix) under periodic conditions of latent (**A**,**C**): non periodic conditions (**B**,**D**), and grids with different number of points, $n_p = 20$ (**A**,**B**)) and $n_p = 100$ (**C**,**D**). (**E-F**): Mask for single points in 2D grids close to latent space boundaries (**E**,**G**) and away from them (**F**,**H**), under non periodic latent space (**E**,**F**) and non periodic latent space (**G**,**H**).

**Synthetic 1D case**  Temperature was set to $\tau = 1$, contrastive regularization strength coefficient to $\bar{c} = [2, 0.5]$. Number of points per dimension of the grid was set to $N = 100$.

Number of batches (each corresponding to a grid) per epoch was set to 100 and models were trained for 10 epochs per each value of $\bar{c}$. Learning rate was set to 0.01. Before being presented to the model neurons, all images where rescaled to have a mean of 0 and a standard deviation of 0.2. Images were generated with a $30 \times 30$ pixel resolution, matching the resolution of the Gabor based neuron models.

**Synthetic 2D case** Temperature was set to $\tau = 1$ in the case of non-periodic latent $\tau = 0.3$ in case of periodic latent (see Appendix C). Constrastive regularization strength coefficients were set to $\bar{c} = [1, 0.5]$. Number of points per grid dimension was set to 20, resulting in grids of 400 points. Learning rate was set to 0.01, number of batches (each corresponding to a grid) per epoch was set to 100 and models were trained for 100 or 120 epochs per each value of $\bar{c}$ respectively in the case of non periodic latent space and periodic latent space. Nonetheless, CPPNs appeared to converge much faster for each regularization strength coefficient considered (20 epochs being sufficient). Before being presented to the model neurons all images where rescaled to have a mean of 0 and a standard deviation of 0.2. Images were generated with a $30 \times 30$ pixel resolution, matching the resolution of the Gabor based neuron models.

**Macaque complex cell** Temperature was set to $\tau = 1$, contrastive regularization strength coefficient to $\bar{c} = [1, 0.5]$. Number of points per dimension of the grid was set to $N = 20$. Number of batches (each corresponding to a grid) per epoch was set to 50 and models were trained for 10 and 20 epochs per each value of $\bar{c}$ in the 1D and 2D case, respectively. Learning rate was set to 0.05 and 0.01 in 1D case and 2D case, respectively. Before being presented to the ensemble model all images where rescaled to have a mean 0.2019 (corresponding to the mid grayscale value of the images on which the ANNs in the ensemble model were fitted), to have a standard deviation 0.15, and if necessary pixel values were clipped to the values corresponding to the extremes of the grayscale on the images on which the macaque model was fitted $[2.1919, -1.7876]$. The standard deviation value was selected to allow clear identification of maximally exciting features while avoiding excessive clipping.

### A.2. Pixel-based MEI optimization

In this method, the pixels of an image are defined as learnable parameters and are learned via gradient-based optimization such that the activation of a target neuron is maximized. Specifically, we defined an input image of size $93 \times 93$ for monkey V1 neuron and $30 \times 30$ for Gabor based neurons and used the Adam optimizer with learning rate of 0.01 to obtain an MEI after 2000 training steps.

### A.3. Software and hardware specifications

All code for model definition, training, evaluation and experiment tracking were implemented in Python 3.9 using PyTorch (Paszke et al., 2019), NumPy (Harris et al., 2020), Weights & Biases (Biewald, 2020), and Docker (Merkel, 2014) packages. All CPPN models were trained using the Adam (Kingma and Ba, 2014) optimizer on a Tesla V100-SXM2-32GB GPU, and took a few minutes to train.

## Appendix B. Gabor-based model neurons

On synthetic data, we tested our approach on a simple cell, a complex cell, an orientation-invariant neuron and two phase-and-orientation-invariant neurons.

- The simple cell was implemented as a Gabor filter followed by ReLU nonlinearity.

- The complex cell was implemented as an energy model (Adelson and Bergen, 1985).

- The orientation-invariant neuron was implemented as a set of simple cells with different orientations, followed by a max-pooling operation.

- The phase-and-orientation-invariant neurons were implemented as sets of complex cells with different orientations, followed by a max-pooling operation.

## Appendix C. The effect of temperature on invariance manifold learning

Orientation-invariant neuron           Complex cell

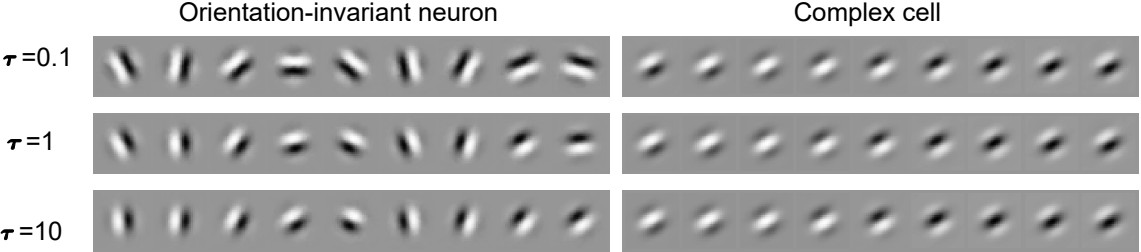

Fig. S2: Images in invariance manifolds corresponding to equally spaced points in 1D non periodic latent space for different temperature values (row) and invariances (columns)

Different choice of temperature can have a important effect on the extent of the invariance manifold that is learned. Fig. S2 shows how in the case of a non-periodic 1D latent space the extent depends as well on the type of invariance of the model neuron. This is due to the fact that images belonging to different invariant transformations have different similarity values and because the temperature parameter controls how strongly to encourage and penalize similarity to positive samples and negative samples, respectively.

In the case of 1D non-periodic latent space, the invariance manifold corresponding to an orientation invariant odd Gabor neuron (Fig. S2 left column) ranges from being almost complete transformation (for low temperatures) to be slightly more than half (high temperature). The phase invariance of complex cell (Fig. S2 right column), on the contrary, is learned up to half a transformation (180 degrees) for all the temperature values considered.

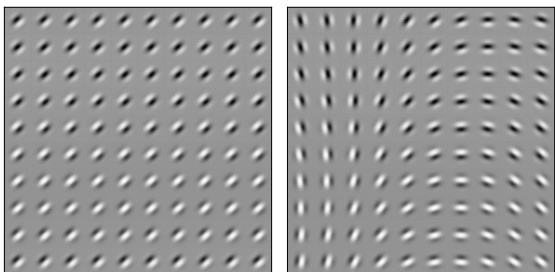

Fig. S3: Effect on temperature on learning 2D invariances. Both figures correspond to an phase-and-orientation-invariant neuron whose invariances are learned with a CPPN with 2D periodic latent space, but with different temperature, respectively $\tau = 10$ and $\tau = 0.3$ from left to right.

When a neuron presents multiple invariances, temperature has an effect also on *which* invariance transformations are learned. See Fig. S3 as example. For a phase-and-orientation-

invariant neuron, a CPPN mapping from a 2D periodic latent space is able to capture orientation invariance only at low temperatures. At high temperature only phase invariance is learned and one of the latent space axes is ignored. This results shows how the optimization objective can, for specific invariances and temperature values, be maximized in the scenario in which the CPPN selectively learns only one of the invariances.

## Appendix D. Learning 2D invariances with 1D latent space

In this appendix we assess the outcome of our method when the invariance manifold of the neuron considered is higher dimensional than the latent space from which the CPPN maps. For this purpose we considered the scenario in which a CPPN mapping from a 1D periodic latent space is trained to identify the invariance of a phase-and-rotation-invariant neuron (2D invariance manifold). We performed the same experiment for high temperature ($\tau = 10$), low temperatures ($\tau = 0.3$) and multiple seeds. With the exception of temperature, training details match the ones reported for 1D synthetic case. Results are shown in Fig. S4.

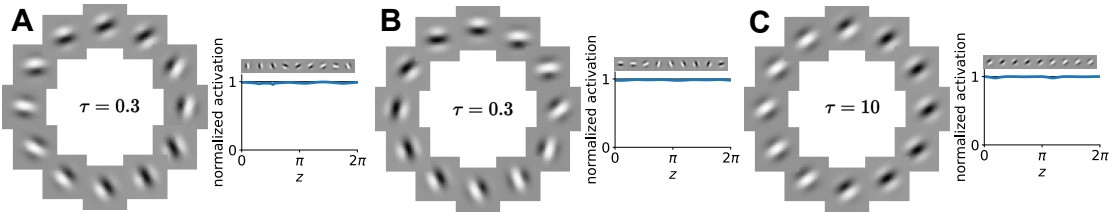

Fig. S4: A CPPN mapping from a 1D periodic latent space is trained to capture the invariance manifold of a phase-and-rotation-invariant-neuron. Images corresponding to equidistant points in latenst space are shown, together with activation. (**A**-**B**) corresponds to different seed, (**B**-**C**) to different values of temperature.

As can be seen from the diversity of images smoothly varying and from the activation being maximized for all images, in all cases considered the CPPN learns a 1D submanifold of the 2D invariance manifold. Fig. S4 further illustrates however how the nature of the learned submanifold might depend on multiple factors such as the nature of the invariances in higher dimensional invariance manifold (e.g. phase vs orientation), the CPPN initialization (inter-seed variability), and training details (e.g., optimization objective). In the case considered, the only invariance learned in the case of high temperature is phase (similarly to what happens in S3). This results shows how the optimization objective can, for specific parameter configurations, be maximized in the scenario in which the CPPN ignores one of the invariances.

### Appendix E.  Approximating a discontinuous invariance manifold with continuous parameterization

The method presented implicitly assumes the MEI invariance manifold to learn to be continuous, as many interesting biological neurons' invariances are smooth. A given neuron could however present a discontinuous invariance. In this appendix we illustrate how the CPPN can address this situation learning to approximate a discrete invariance manifold with a continuous latent space, thanks to the introduction of jumps. Specifically we considered a polarity invariant neuron obtained max pooling the responses of ON and OFF centered even simple cells (same parameters, except for phase). Such neuron presents an invariance manifold that consists in two point in the image space, corresponding to the ON and OFF centered linear filters of the simple cells from which it is composed. We trained to CPPN mapping from a 1D periodic latent space to approximate the discontinuous invariance manifold of such polarity invariant neuron. Temperature was set to $\tau = 0.3$. Remaining training details match the ones reported for the 1D synthetic case in appendix A.

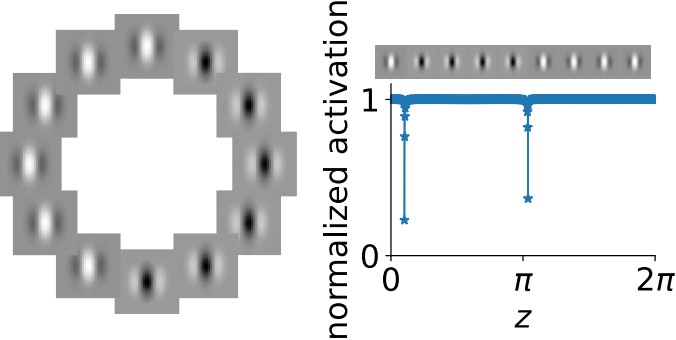

Fig. S5:  Learned invariance manifold and corresponding activations in the case of a CPPN mapping from a 1D periodic latent space and trained to learn the discrete manifold of a polarity invariant neuron MEIs are reported for 12 equally spaced points and activation for 1000 equally spaced points in the latent space. Two domains, each of them corresponding to one of the two MEIs, appear in latent space. The necessity of connecting such domains via a continuous parameterization corresponds to the arising of sudden jumps in image space connecting such domains, during which activation drops. The small fraction of points corresponding to activation sensibly deviating from MEI activation gives a measure of how localized jumps are in latent space.

Fig. S5 demonstrates that CPPN can learn to approximate the two-point invariance manifold mapping large domains of the latent space to the same maximally exciting image and introducing sudden jumps between such domains. This approximate learning of a discontinuous and discrete manifold with a continuous latent space is possible thanks to the decreasing strength of the contrastive objective during training. In the first part of the training, the high regularization strength $c$ of the contrastive objective tends to be predominant and the CPPN learns a smooth manifold of images that are particularly diverse, that

tend to highly activate the neuron but that are not exactly maximally exciting. When the regularization strength decreases, however, the activation objective becomes predominant over the contrastive objective that ensures smoothness and the CPPN learns to introduce jumps between the domains in which the generated images look the same.

## Appendix F. Periodic 2D latent space on cylinder invariance topology

This appendix displays some of the results obtained when fitting a 2D periodic manifold on a non-periodic invariance (cylinder topology).

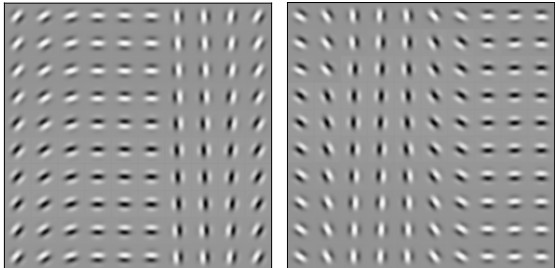

Fig. S6: Two different instantiations (same hyperparameters, different seeds) of the same CPPN learning a phase-and-partially-orientation-invariant neuron invariance manifold with a 2D periodic latent. Fitting a 2D periodic latent on a non-periodic invariance manifold forces either sudden jumps (left figure, jump on the orientation axis) or to learn the same transformation twice (right figure, rotation of 90 degrees is learned first clockwise and than anticlockwise)

## Appendix G.  Activations corresponding to images in Fig. 3

| | simple cell | complex cell | phase and orientation invariant neuron | phase and partially orientation invariant neuron |
|---|---|---|---|---|
| 2D non periodic latent | $1 \pm 8e\text{-}8$ | $0.991 \pm 0.006$ | $0.991 \pm 0.007$ | $0.991 \pm 0.006$ |
| 2D periodic latent | $1 \pm 9e\text{-}8$ | $0.990 \pm 0.007$ | $0.985 \pm 0.007$ | $0.985 \pm 0.006$ |

Table 1: Mean and standard deviation of the activations corresponding to images generated from the learned invariance manifold when using a 2D latent space (corresponding to images shown in Fig. 3).

## Appendix H.  Selection of complex cells

We identified complex cells following these steps:

1. We created an ensemble model, averaging the predictions of $n = 3$ ANNs implemented and trained as described in 3.4.

2. We trained multiple instances of a linearized version (LN models) of the ANN model obtained by simply dropping the nonlinearities in the model (except the last one which ensures positive-values firing rates).

3. We computed a nonlinearity index for each neuron by comparing the correlation of the ensemble model predictions with trial-averaged neural responses with the highest correlation achieved by any of the trained LN models: $I_{nl} = (c_{ens} - \max(c_{lin}, 0))/c_{ens}$.

4. Among neurons with high $I_{nl}$ we selected the ones with $c_{ens} > 0.8$ to consider only nonlinear neurons well modelled by our ensemble.

5. We performed direct pixel optimization to identify one MEI per neuron

6. We selected the neurons whose MEI visually resembled a Gabor filter.

## Appendix I. Analysis of the MEIs generated via CPPN

To better show that our method has captured the previously shown phase invariance in monkey V1 complex cells, for each image generated via the CPPN (Fig. 4**B**–**D**) we learned a Gabor filter that results in the least mean squared error (Bashiri, 2020), and assessed the phase of the learned Gabor filters. Fig. S7 shows that the learned Gabor filters are a close match to the MEIs both qualitatively (i.e. visually) and quantitatively (i.e. resulting activations). Importantly, as we go along the learned invariance manifold the phase of the learned Gabor smoothly changes and the manifold covers the complete $2\pi$ cycle.

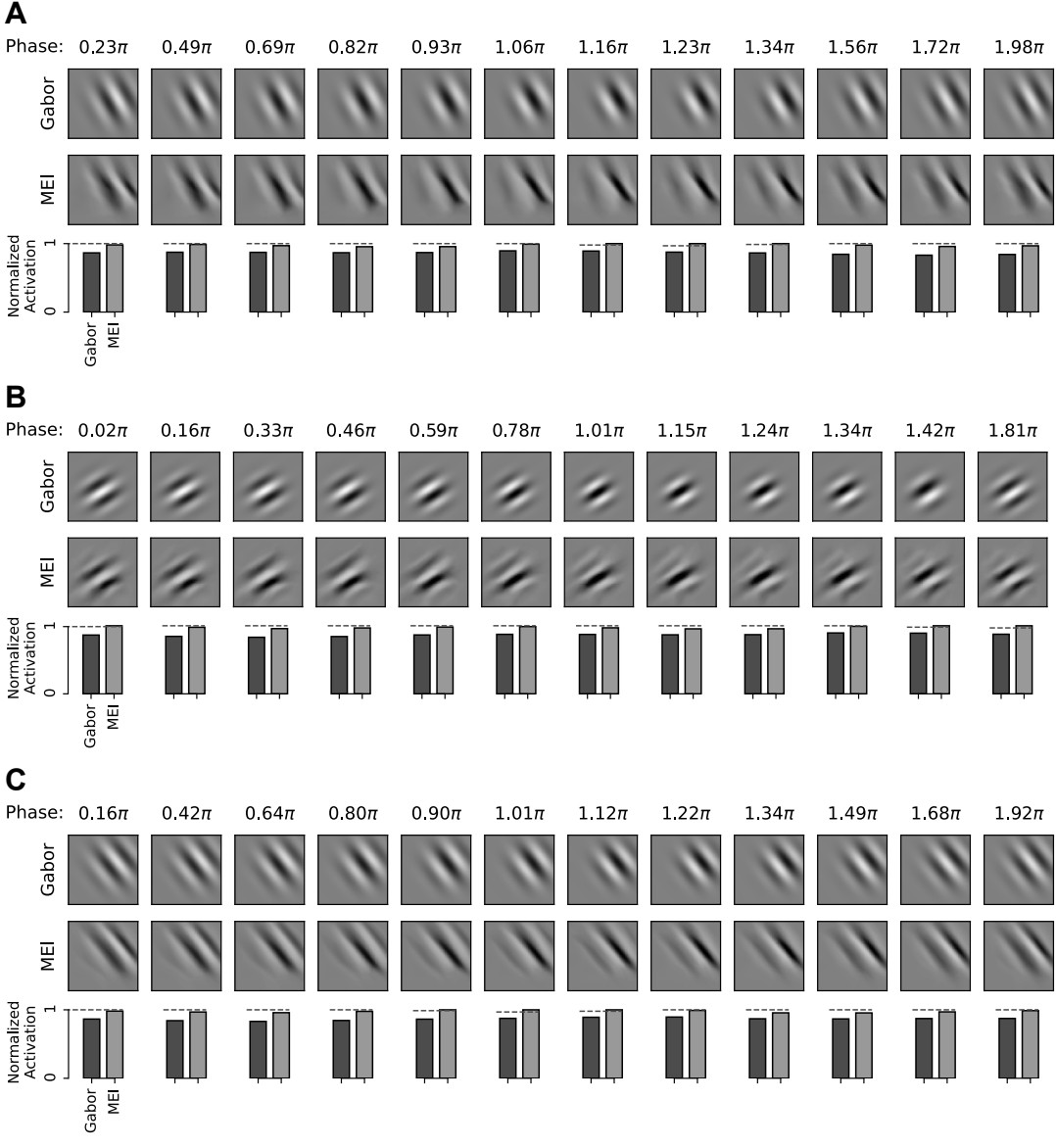

Fig. S7: Analysis of the MEIs generated via CPPN using fitted Gabors. **A**, **B**, and **C** correspond to the neurons shown in Fig. 4B, 4C, and 4D, respectively. For visualization purposes, MEIs and Gabors are cropped around the receptive field of the neurons.

