# OpenReview forum: "Learning Invariance Manifolds of Visual Sensory Neurons"
_NeurIPS.cc/2022/Workshop/NeurReps — NeurReps 2022 Poster_

### Official Review · Reviewer_zcuu · 2022-10-11
**Could use more rigorous experimental analysis, but built on interesting underlying ideas**

**Confidence:** 3
**Soundness:** 3
**Presentation:** 4
**Contribution:** 3
**Overall Rating:** 7

**Summary:**

The paper builds a model to identify features along which sensory neurons are selective to across an array of conditions.

The network maps a latent representation to a manifold in image space, followed by a readout layer with a Poisson loss to predict neural activity.

They use varying topologies to see how well the network disentangles representations.

**Questions:**

-One question that always pops into my mind with contrastive learning is how the positive and negative samples are chosen. I see this is addressed in the appendix, but I'd be interested about the stability of the network with respect to different transformations

-Can you say more about the nonlinearity index in your own words? I'm not familiar with this metric or why it's biologically meaningful. Justification for this could be clearer

-Why is this a hard problem? What is the "control" network or naive model whose problems your architecture is able to overcome?

**Limitations:**

I get the visual appeal of Figure 3, but is there a more quantitative way that you can show this result? I think the main limitation is that the presented experiments are light. Why not use the ground truth for more numerical results?

What are the baselines of comparison? What if you removed different aspects of the loss function? How would your error change?

**Recommended Decision:**

3: Accept

**Relevance:**

4: Highly relevant

**Strengths And Weaknesses:**

-Originality: Though I'm not familiar with this space, the authors present a well-written related work section to explain the importance of the problem and the originality of their manifold approach. I'm not entirely sold on the "data-driven" argument made in the introduction: we still must choose which stimuli we choose to present, for example (it's a minor concern, nonetheless)

-Quality: The network is described in great detail and the biological motivation is set up well.

-Clarity: The paper is well-written, polished, and professional. However, as expanded upon below, the experimental results could use more discussion. It is currently unclear to me why Figure 4 (esp A) supports the claims made in the conclusion.

-Significance: The main experimental results are presented in FIgs 3 and 4. Though the architecture seems promising, these results seem somewhat limited in their rigor and breadth.

**Submission Track:**

Proceedings Paper (9 Page)

---

> ### Author Response · Authors · 2022-11-02
> **Authors response to Reviewer zcuu (part 2)**
>
> **Re: Nonlinearity Index**: We used this empirical metric, previously published in [2], to select complex cells which can be well characterized by nonlinear models, but are not well characterized by linear models . The formula can be seen in appendix E, and the outcome of such a measure can be seen in Figure 4A. The measure is based on the comparison between the prediction performance obtained by a linear model (or LN) and the performance of a nonlinear one. Complex neurons yield a high nonlinearity index (further from the diagonal) compared to simple cells (close to the diagonal).  This method is one way to identify such cells among other approaches (e.g. STA) and is purely computationally motivated and is not directly inspired by biological considerations.
>
> **Re: why is this a hard problem?**: Figuring out what is the nature of sensory representation in higher visual areas is a hard problem and we, as a field, are still far from solving it. Here we propose a novel tool that has the potential to help solving this problem, as invariances are a key aspect of sensory representations. Previous approaches yield a discretized version of the invariance manifold (single images) where the resulting images are not necessarily ordered and results are therefore harder to interpret. Our proposed method is, to the best of our knowledge, the first method that attempts to obtain an explicit topological mapping of the invariance manifold, which was achieved by using the contrastive learning objective combined with the CPPN. Briefly, the contrastive objective yields ordered generated images and the CPPN results in a continuous parameterization of the manifold.
>
> **Re: more quantitative way of showing results in Fig 3**: See response to **Re: results seem somewhat limited in their rigor and breadth**.
>
> **Re: Baseline of comparison and analysis of loss function**: To assess whether the If we understand this correctly, in our case baseline is the activation corresponding to one MEI, obtained via direct pixel optimization to maximize the response of a given neuron. We have explored the contribution of each term in the objective function in the past:
> - If we remove the contrastive learning objective, the CPPN learns to generate maximally exciting images (MEIs), but it is not encouraged to diversify the generated images.
> - If we remove the activation term in the objective function, the CPPN generates images that are smoothly varying but do not contain any relevant information regarding the neuron function as neuron model output is not used at all in the objective.
> These results correspond to the contribution of each term in the objective function as explained in the section 3.2 of the manuscript.
>
> [1] Walker, Edgar Y., et al. "Inception loops discover what excites neurons most using deep predictive models." Nature neuroscience 22.12 (2019): 2060-2065.
>
> [2] Antolík, Ján, et al. "Model constrained by visual hierarchy improves prediction of neural responses to natural scenes." PLoS computational biology 12.6 (2016): e1004927.

---

> ### Author Response · Authors · 2022-11-02
> **Authors response to Reviewer zcuu (part 1)**
>
> Thank you very much for your positive feedback. Below is a detailed response to the points you raise in your review, as well as how we plan to improve the final version of the manuscript accordingly.
>
> **Re: the “data-driven” argument in the introduction**:  The method is “data-driven” in the sense that the invariance dimensions in the image space are completely identified using the data (i.e. neuronal responses). That is, there is no need to hypothesize a specific invariance direction (e.g. phase) and then present stimuli along that direction to assess invariance but rather we let the data inform us about the invariance directions in the stimulus space. Of course this cannot be directly done with real neurons in experimental settings. Therefore, we need a powerful predictive model of neuronal response and for that we train an ANN (in itself obviously a data-driven method as well) to predict responses of neurons to natural images, but this ANN could be replaced by any model that predicts well neural responses (and is differentiable). As with many data-driven methods, the presented method does come with some metaparameters, for example the dimensionality of the latent space.
>
> **Re: Figure 4 (esp A)**: Figure 4A shows the nonlinearity index (NI) of neurons. You are right that this figure on its own does not support the claim and it was not intended as such. It rather illustrates the analysis we performed to choose the neurons on which we applied our method to identify their invariance(s). More specifically, this figure shows how neurons were selected to identify complex cells (details in dedicated appendix) which would correspond to neurons with a high NI as opposed to simple cells (with no invariance) which would have a low NI. Once we selected the complex cells using this measure, we then applied our method on these cells to showcase that our method can indeed identify invariances for these cells (Fig 4B-E).To make the purpose of Figure 4A clearer we will modify the figure to indicate which neurons are displayed in Figures 4B-E.
>
> **Re: results seem somewhat limited in their rigor and breadth**: Our main goal was to show that our method can successfully learn directions in the stimulus space that a neuron is invariant to. Assuming that such a manifold exists, this can be assessed by 1) making sure that the images along the learned manifold are diverse and 2) the resulting neuronal activation from these images are not only maximized but also very similar to each other. For (1) a visual inspection of the images would suffice to detect the diversity of images and for (2) we show the resulting activations from the generated images oscillated around 1 after normalization by the MEI activation (obtained using direct pixel optimization similar to the method presented in [1]). This shows that the images generated via CPPN are maximally exciting and the fact that the activations do not change much across different images shows that they are indeed on the invariance manifold. Taken together these results can show whether the method has successfully identified the invariance manifold of maximally exciting images (or at least a submanifold of it). For the results shown in Figures 2 and 4, both of these analyses are present. However, as you rightly pointed out, analysis (2) is not shown for the results in Figure 3. We agree that adding such an analysis would make the claim stronger. Therefore, we will add analysis (2) as an appendix (G) in the camera-ready version.
>
> **Re: choosing positive and negative samples for contrastive learning**: We implemented the contrastive learning objective such that how positive and negative samples are chosen could be configured, both in terms of the shape and size of the neighboring area (the area in which positive samples fall in). We tried different values of positive neighbor region size as well as different positive neighbor region shapes in 2d case (square or circle). In the experiments performed, we obtained qualitatively similar results for different values of both size and shape of the neighboring area. It is worth noting that our experiment was not extensive and the values we tried covered a narrow range. We agree that such an analysis would be insightful and could even result in a better contrastive objective, but an exhaustive exploration would be computationally very expensive.

---

### Official Review · Reviewer_QG7y · 2022-10-11
**Review of “Learning invariance manifolds of visual sensory neurons”**

**Confidence:** 4
**Soundness:** 3
**Presentation:** 3
**Contribution:** 3
**Overall Rating:** 5

**Summary:**

This paper proposes a method to identify the invariance properties of visual sensory neurons by learning a manifold of images that excite them maximally (known as maximally exciting images, or MEIs). This approach extends recent works that optimize MEIs on a pixel-by-pixel basis by learning a differentiable parameterization of images based on CPPNs. The authors demonstrate that this method can be successfully applied to artificial networks with known invariances, as well as to recordings of complex cells from macaque V1.

**Questions:**

1. The proposed method requires the experimenter to manually specify the dimensionality and topology of the latent space. The authors demonstrate that their method can learn to ignore one dimension in a two-dimensional CPPN latent space when the true "invariance manifold" is one-dimensional. Does the method break down if the latent space dimensionality is too large? Similarly, what happens if the latent space is lower-dimensional than the true "invariance manifold"? Developing an understanding of how the method behaves in these scenarios would seem to be an important prerequisite to applying it to cases in which one does not have a good *a priori* estimate of the true dimensionality.
2. In previous works on MEIs, the fitting method was benchmarked by comparing neural responses to the putative MEI to responses to other stimuli, such as Gabor patches. It could be useful to include a similar comparison in the present work, as features such as the approximate phase-invariance of complex cells were originally identified with such hand-designed stimuli.
3. How does the method behave when applied to a system that is invariant under the action of a non-compact group, such as translations?
4. To the best of my understanding, the method assumes a continuous symmetry. Can it successfully discover discrete symmetries?
5. In Figure 4, some small fluctuations in the normalized responses are visible. How large are these fluctuations relative to trial-to-trial response visibility? So far as I can see, there are no error bars.

**Limitations:**

The authors do not provide extensive discussion of the limitations of their method. The question of how strong a limitation is the requirement of specifying latent space structure by hand is in my view probably the most salient issue.

**Recommended Decision:**

2: Borderline

**Relevance:**

4: Highly relevant

**Strengths And Weaknesses:**

I think that this work is an interesting extension of previous works on MEIs, and is, to the best of my knowledge, reasonably novel and technically sound. The idea of learning the invariance properties of neural codes should be highly relevant to the NeurReps community. I describe what I view as its primary weaknesses under **Questions**.

I found the submitted manuscript to be generally clearly written, and the figures are easy to interpret. However, some proofreading is required. For example, in-text figure references in the current manuscript are all to Figure 4 and Figure C (which does not appear to exist, even in the Appendix).

**Submission Track:**

Proceedings Paper (9 Page)

---

> ### Author Response · Authors · 2022-11-02
> **Authors response to Reviewer QG7y (part 2)**
>
> **Re: can the method successfully discover discrete symmetries?**: Our method implicitly assumes the invariance manifold to parameterize to be continuous. However, as we have shown in Fig 3 (right column, bottom row) our method can indeed learn jumps in the image space approximating discontinuities in the invariance manifold. How well our method can approximate discontinuities depends on the topology of the latent space and the contrastive objective. At high regularization strength $c$ of the contrastive objective tends to be predominant and the CPPN learns a smooth manifold of images that highly activates the neuron but are not exactly maximally exciting. When the regularization strengths decrease, however, the activation objective becomes predominant over the contrastive objective ensuring smoothness and the CPPN learns to introduce "jumps" between domains in which the generated images look the same. To further highlight such behavior, we will add an additional appendix (E) in which we show how our method can capture a discontinuous invariance manifold consisting of two points.
>
> **Re: specifying latent space structure by hand is the most salient issue**: While we agree that this is a shortcoming, the method can still learn a meaningful invariance manifold even for the “wrong” choice of the latent space topology, as shown in Figure 3 and Appendices D, E, and F. One way to use the current method for finding the underlying manifold is to experiment with different topologies and make informed choices about the topology of the latent space based on the results of the experiments. However, to further improve the method, we are exploring ways of making the choice of latent space topology data-driven and learnable.

---

> ### Author Response · Authors · 2022-11-02
> **Authors response to Reviewer QG7y (part 1)**
>
> Thank you very much for your helpful and constructive feedback. Below we provide some explanations to address your questions and describe how we plan to improve the manuscript accordingly.
>
> **Re: Some proofreading is required.**:  We appreciate you pointing out the inconsistencies in the original manuscript. We have gone through the manuscript and fixed them.
>
> **Re: does the method break down when the latent space dimensionality is too large?**: We agree that such an analysis would be insightful and would complement the results shown in Figure 3 (two left columns). However, for the present proof-of-concept study, we have not performed this analysis for the following reasons:
> - Computationally expensive: the number of images generated at each training step (corresponding to the number of grid points in the latent space) grows exponentially with the number of dimensions in the latent space. That is, for a $d$-dimensional latent space with $n$ points per dimensions we will have $n^d$ generated images. Therefore, using high dimensional latent spaces can become computationally expensive. To solve this problem, we would need a different strategy to explore the latent space, which we have not pursued yet.
> - Hard to visualize: visual inspection of the images generated from a high dimensional latent space is challenging. Here we used 1D and 2D latent spaces to easily visualize the results and facilitate interpretation of the invariance transformations. For our proof-of-concept we thus focussed on low dimensional latent spaces.
>
> **Re: what happens if the latent space is lower-dimensional than the true "invariance manifold"?**: Our results show that the CPPN learns a submanifold of the true invariance manifold. The learned transformation (corresponding to the submanifold) might however depend on multiple factors such as the true invariance manifold, the CPPN initialization, and training details (e.g., the optimization objective, etc.). For the camera-ready version we will add complementary results to appendix (D) to address this comment. Specifically, we will use a 1D latent space for a neuron model with 2D invariance manifold and show the resulting images as well the activations.
>
> **Re: benchmarking by comparing responses to putative MEI to other stimuli such as Gabor**: If we understood correctly, the aim of such an analysis would be to show that highly exciting images are more complex than simple gabor filters (similar to the analysis performed in Walker et. al 2019). Although this could still be an interesting analysis, it was not a point we wanted to make in this work. Furthermore, such an analysis would require in-vivo experiments which was out of the scope for the current study. Our goal was to demonstrate that our method indeed finds familiar previously shown invariances such as phase as is expected from complex cells in monkey V1. Nonetheless, the analysis you are suggesting can complement the results for real neurons (Figure 4) in two ways:
> - It could show that the visual pattern of the generated images are very much Gabor-like
> - It could help us show that what our method has learned is indeed phase-invariance
>
> Therefore, for the camera-ready version, we will add additional results to the appendix (I) illustrating these points. More specifically, for each image we will find the Gabor filters that results in the least mean squared error and would then show:
> - The activation that the Gabor filters results in
> - How the phase changes for different Gabor filters corresponding to different generated images along the invariance manifold
>
> **Re: how does the method behave when applied to a system that is invariant under the action of a non-compact group, such as translations?**: As neurons are usually driven by contrast and, we limit the contrast of the MEI to prevent activation from diverging and to allow fair activation comparisons across images. This means that the entire manifold of MEIs is enclosed in an n-sphere in image space, which is a compact space. Therefore, the manifold itself must be compact too.

---

### Official Review · Reviewer_wgF1 · 2022-10-11
**A neat idea but needs proper testing**

**Confidence:** 4
**Soundness:** 2
**Presentation:** 1
**Contribution:** 2
**Overall Rating:** 4

**Summary:**

The authors try to find a manifold of images that highly activate synthetic and real neurons. They use a pretrained image to neuron model, then attach a xy (+z) to image moel at the front, and use the latter model to find images that most activate a single neuron. Critically, xy runs over the image dimensions, and they choose z to come from a predefined manifold. They then encourage the model to produce a diverse set of images. In the synthetic neuron case they learn images from an invariance manifold. In the real neuron case it is less clear.

**Questions:**

See limitations

**Limitations:**

The big one is whether this really work on real neurons not just synthetic ones.

In Fig. 4 you present results on real data. To make it convincing, rather than just showing images, you would need to extract phase / rotation from the images and then present those numbers. The point you're making isn't clear from the images.

There is also a big bias towards learning a smoothly changing rerpresentations. To make the claim about a partivular neuron caring only about a invariant manifold, you would also need to check no other images don't highly activaste this neurons. You could do this with z being a one-hot code, rather than something smoothly varying.

**Recommended Decision:**

2: Borderline

**Relevance:**

4: Highly relevant

**Strengths And Weaknesses:**

Originality: This is fun and original.
Quality: The idea is interesting, but it's a long way from being useful at the moment. Needs much more thorough testing.
Clarity: This is not a well written paper. Many modelling details / choices not explained.
Significance: If this could be a general method for many neurons across the brain then it would be very exciting.

**Submission Track:**

Proceedings Paper (9 Page)

---

> ### Author Response · Authors · 2022-11-02
> **Authors response to Reviewer wgF1 (part 2)**
>
> **Re: bias towards learning a smoothly changing representations**: You are right, and that design choice was on purpose since many interesting invariances are smooth. However, even though it assumes a continuous manifold it can learn fast jumps as we have shown in Figure 3 (right column, bottom row). How well our method can approximate discontinuities depends on the topology of the latent space and the contrastive objective. At high regularization strength $c$ of the contrastive objective tends to be predominant and the CPPN learns a smooth manifold of images that highly activates the neuron but are not exactly maximally exciting. When the regularization strengths decrease, however, the activation objective becomes predominant over the contrastive objective ensuring smoothness and the CPPN learns to introduce jumps between domains in which the generated images look the same. We will add an appendix (E) in the camera-ready version showing that our method can indeed approximate a discontinuous manifold.
>
> **Re: checking whether other images highly activate a target neuron**: We do not claim that the proposed method captures the entirety of the invariance manifold, but it can successfully learn at least parts of the underlying manifold. In our experiments we indeed had such an observation, as we have shown in Figure 3 and in Appendix C. We agree that assessing how well our method captures the complete invariance manifold is an important analysis. Regarding your suggestion of using a one-hot encoding instead of a continuous latent variable, the main issue is that this approach would result in a discretize characterization of the invariance manifold, exactly the limitation our method was designed to rectify, requiring an a priori specification of number of MEIs to be generated, which is very similar to previous methods [1, 2]. However, it would be interesting to use multiple latent spaces to cover the complete invariance manifold even in the presence of discontinuities. We plan to explore such directions in the future.
>
> [1] Walker, Edgar Y., et al. "Inception loops discover what excites neurons most using deep predictive models." Nature neuroscience 22.12 (2019): 2060-2065.
>
> [2] Cadena, Santiago A., et al. "Diverse feature visualizations reveal invariances in early layers of deep neural networks." Proceedings of the European Conference on Computer Vision (ECCV). 2018.

---

> ### Author Response · Authors · 2022-11-02
> **Authors response to Reviewer wgF1 (part 1)**
>
> Thank you very much for your feedback. We plan to improve the final version of our manuscript with the addition of several analyses addressing some of your concerns. Below is a detailed response to the points raised in your review:
>
> **Re: for real neuron case results are less clear**: Our main goal was to show that our method can successfully learn directions in the stimulus space that a neuron is invariant to. Assuming that such a manifold exists, this can be assessed by 1) making sure that the images along the learned manifold are diverse and 2) the resulting neuronal activation from these images is not only maximized but also very similar to each other. While a more thorough analysis would be possible for the toy examples since the ground truth is known, it might fail for real neurons where the invariances can be more complex. In the case of the real neurons, the most important aspect is that the activity stays high along the manifold. Therefore, we decided to quantify the activation and to stay with a qualitative inspection for the images, both of which are illustrated in Figure 4.
>
> **Re: many modeling details / choices not explained**: Thank you for pointing this out. While we aimed to provide as many modeling details as possible and explain their choices in section 3 as well as Appendix A, we realized that a more complete description can be provided adding more details and explanation. Below is a list of details we will add to the camera-ready manuscript:
> - Details of pixel-based MEI optimization
> - Why we used 1- and 2-dimensional latent space
> - Details of the dataset
>     - Preprocessing of the images (rescaled, etc.)
>     - Size of the image shown to the ANN model
>      - Size of Gabor filters for the toy data
>
> If beyond these aspects you still find that certain details are missing, we would appreciate it if you could point them out, so that we can provide all the details necessary in the manuscript (or here in the responses if the manuscript cannot be edited further).
>
> **Re: method applied on many neurons would be very exciting**: We completely agree that applying this method on many neurons or neurons from higher visual areas would potentially yield very interesting results. However, for this manuscript, we focused on providing a proof-of-concept and demonstrated our method on previously shown results (e.g. phase invariance of complex cells). Applying our method to many neurons is one of the future directions we are excited about.
>
> **Re: results in Fig. 4 are not convincing. Need to extract phase/rotation**: Based on our response to **Re: for real neuron case results are less clear**, we hoped to convince the reader that the method has successfully learned the invariance of real neurons. However, we agree that it might be difficult to visually assess the diversity of the generated images due to the higher spatial frequency. Displaying the manifold as a video (as we do internally) instead of a set of images would have resulted in a more clear illustration of the identified variance, but this is unfortunately not possible in the present proceedings format. While in general it is not trivial to assign a name to the invariance of a neuron, it is visually apparent from the images that if we want to assign a name to the learned invariances it would be phase, as is expected from complex cells in monkey V1. Nonetheless, to facilitate the clarity of the results and to better show the success of the method in finding diverse images, for the camera-ready version, we will add as an appendix (I) an additional analysis of the generated images to show the change in phase as we move along the invariance manifold. More specifically, for each image we will find the Gabor filter that results in the least mean squared error and would then show:
> - The activation that the Gabor filters results in
> - How the phase changes for different Gabor filters corresponding to different generated images along the invariance manifold

---

### Decision · Program_Chairs · 2022-10-21

Accept (Poster)